# TAMING FLOW-BASED I2V MODELS FOR CREATIVE VIDEO EDITING

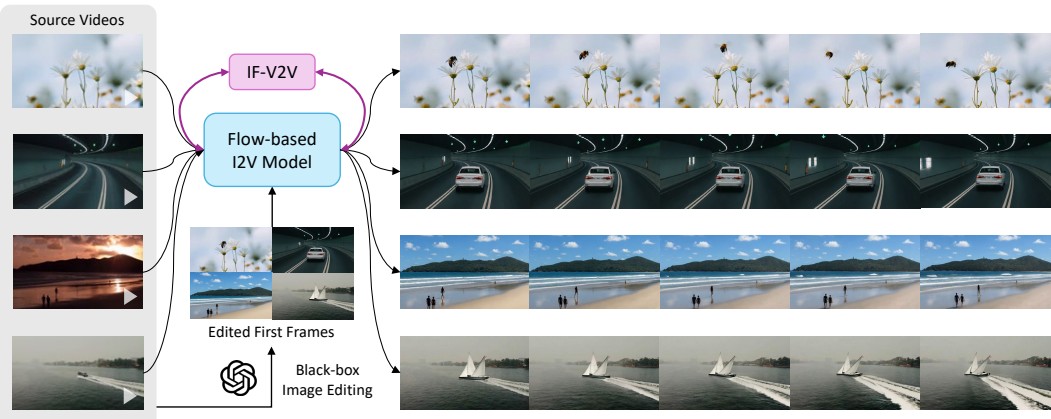

Figure 1: Illustration of IF-V2V, a lightweight plug-and-play method for creative video editing (§1). It effectively combines the capability of black-box image editing approaches and flow-matching-based I2V models without inversion and optimization, achieving various creative editing tasks with high visual quality.

## ABSTRACT

Although image editing techniques have advanced significantly, video editing, which aims to manipulate videos according to user intent, remains an emerging challenge. Most existing image-conditioned video editing methods either require inversion with model-specific design or need extensive optimization, limiting their capability of leveraging up-to-date image-to-video (I2V) models to transfer the editing capability of image editing models to the video domain. To this end, we propose IF-V2V, an Inversion-Free method that can adapt off-the-shelf flow-matching-based I2V models for video editing without significant computational overhead. To circumvent inversion, we devise Vector Field Rectification with Sample Deviation to incorporate information from the source video into the denoising process by introducing a deviation term into the denoising vector field. To further ensure consistency with the source video in a model-agnostic way, we introduce Structure-and-Motion-Preserving Initialization to generate motion-aware temporally correlated noise with structural information embedded. We also present a Deviation Caching mechanism to minimize the additional computational cost for denoising vector rectification without significantly impacting editing quality. Evaluations demonstrate that our method achieves superior editing quality and consistency over existing approaches, offering a lightweight plug-and-play solution to realize visual creativity.

## 1 INTRODUCTION

Visual content editing aims at manipulating images and videos to align with the user's intention, offering endless possibilities in film production and creativity (Zhang et al., 2025). Although significant progress has been made in the realm of image (Huang et al., 2024; Yu et al., 2025; Han et al.,

2024; Zhao et al., 2024; Xiao et al., 2024; Shi et al., 2024b; Mao et al., 2025; Liu et al., 2025b; OpenAI, 2025), video editing is still in its infancy due to the difficulty of maintaining spatiotemporal consistency, the lack of massive training data, and the huge computational cost (Sun et al., 2024). Thus, transferring the strong editing capability of image editing models to the video domain by image-conditioned video editing serves as an ideal choice for creators to implement their ideas.

Most existing image-conditioned video editing methods either rely on the inversion of the diffusion process (Song et al., 2021; Karras et al., 2022) or require extensive optimization. The inversion process not only introduces a significant computational burden but is also inherently inaccurate (Sun et al., 2024). To compensate for such error, a series of strategies have been introduced to enhance the texture and motion consistency, such as attention map manipulation (Ouyang et al., 2024) and motion embedding optimization (Song et al., 2025). Despite their effectiveness, these strategies are tailored for specific models, lacking the universality to adapt to other image-to-video (I2V) models. Optimizing either latents or model parameters (Yan et al., 2023a; Shi et al., 2024a; Yan et al., 2023b; Jiang et al., 2025; Jeong et al., 2025) requires extensive computational resources or data, which is not friendly for common users. In addition, it also lacks the flexibility to switch between various I2V models. With the rapid emergence of powerful flow-matching-based I2V models with different DiT-based architectures (Wan et al., 2025; Xu et al., 2024a; Kong et al., 2025; Yang et al., 2025; Peng et al., 2025; Fan et al., 2025a; Lipman et al., 2023; Liu et al., 2022; Do et al., 2025; Peebles & Xie, 2023), a model-agnostic optimization-free editing paradigm is supposed to be promising to fully unleash the strong prior of these models with billions of parameters.

We introduce IF-V2V, an Inversion-Free image-conditioned video editing method that can be applied to off-the-shelf flow-matching-based I2V models within acceptable computational overhead (Fig. 1). It allows users to flexibly combine the capability of any black-box image editing methods and semi-black-box flow-matching-based I2V models with access to their input latents and denoising vectors. This paper primarily encompasses the following three technical contributions: **First**, to incorporate source video information into the denoising process without inversion, we introduce Vector Field Rectification with Sample Deviation (VFR-SD). This method modifies the vector field used in solving the target ordinary differential equation (ODE) by adding a deviation term. Specifically, this deviation term leverages the difference between the ground truth sample and the predicted expectation of the source video distribution to direct the target denoising path to align with the source video sample. **Second**, to further enhance spatiotemporal consistency with the source video, we present Structure-and-Motion-Preserving Initialization (SMPI), which utilizes the motion cue of the source video to generate temporally correlated noise for initialization and meanwhile embeds the structural information into ODE initializations and reference conditions. **Third**, to minimize the additional computational cost for vector field rectification, we devise a Deviation Caching (D-Cache) mechanism to reuse the deviation term while preserving editing quality according to the variation pattern of the target denoising vector (Liu et al., 2025a).

Extensive experiments demonstrate that IF-V2V achieves superior visual quality and consistency in image-conditioned video editing tasks with modest additional computational cost. Our method also outperforms previous approaches across diverse editing paradigms consistently. Thanks to the model-agnostic design, IF-V2V can effectively combine the capability of any state-of-the-art image editing and I2V models to support a variety of creative video editing tasks, demonstrating a strong potential to serve as a lightweight solution for creators to experiment with their innovative ideas.

## 2 RELATED WORK

**Image-to-video Generation.** Visual content generation and editing have witnessed significant advancements thanks to the emergence of diffusion models (Ho et al., 2020; Song et al., 2021; Rombach et al., 2022). Recently, DiT (Peebles & Xie, 2023) has become the mainstream architecture of the denoising model with promising generation quality, surpassing U-Net (Ronneberger et al., 2015) with its powerful scaling capability (Kaplan et al., 2020) and potential for multimodal interaction (Esser et al., 2024). Flow Matching (Lipman et al., 2023; Liu et al., 2022) introduces an improved generative model paradigm that interpolates data and noise linearly in the forward diffusion process, bringing better theoretical properties and conceptual simplicity. Building upon these works, a number of I2V models (Wan et al., 2025; Xu et al., 2024a; Kong et al., 2025; Yang et al., 2025; Peng et al., 2025; Fan et al., 2025a) have emerged with full 3D attention (Vaswani et al., 2017)

instead of decoupled spatiotemporal attention (Guo et al., 2024), significantly enhancing generation quality and consistency.

**Training-free Visual Editing.** Training-free visual editing modifies the source image or video according to designated conditions (e.g., text, image, and mask) at test time, using off-the-shelf pretrained models. Existing works can be broadly categorized into two categories: inversion-based and optimization-based methods. Inversion-based methods (Fan et al., 2025b; Yoon et al., 2025; Feng et al., 2025; Yatim et al., 2025) adopt the inversion of the diffusion process to map the input back to Gaussian noise, and then perform denoising under given conditions. However, not only is the inversion process time-consuming, but it also inevitably induces error. To overcome the inherent inaccuracy of inversion and ensure consistency with the input, various attention injection strategies (Feng et al., 2025; Yatim et al., 2025) are utilized to further incorporate source information. Despite their effectiveness, these strategies are model-specific, reducing their universality to different model structures. Optimization-based methods (Jeong et al., 2025; Ren et al., 2025; Gao et al., 2025) use SDS (Poole et al., 2023) to directly optimize the input latents towards the desired direction. Nevertheless, the optimization operation introduces considerable computational cost, limiting its availability to common creators. With the prevalence of flow-based models (Lipman et al., 2023; Liu et al., 2022), there have also been methods (Avrahami et al., 2025; Dalva et al., 2024; Xu et al., 2025) that leverage the properties of the flow matching process to achieve more precise and consistent visual editing. However, few solutions are both lightweight and universal without model-specific design in the video domain, limiting creators to swiftly leverage the most up-to-date I2V base models for video editing within user-friendly resources, such as a single GPU.

There have also been works exploring inversion-free image editing. For instance, InfEdit (Xu et al., 2024b) theoretically depends on the diffusion process, limiting its application to state-of-the-art flow-based models. It also needs attention manipulation, further limiting its universality. FlowEdit (Kulikov et al., 2024) leverages flow properties to construct a transport from the source to the target distribution, which is derived from the Euler Discrete Solver (Esser et al., 2024). In contrast, our method constructs two parallel ODEs to model the editing process, which does not depend on a specific ODE solver and enables control over editing strength. Our method also introduces SMPI to further enhance video-level spatiotemporal consistency and a flexible caching strategy.

## 3 METHODOLOGY

### 3.1 TASK FORMULATION

Given a source video $x^{src} = \{x_i^{src}\}_{i=1}^{L}$ with $L$ frames and the edited first frame $x_1^{edit}$, image-conditioned video editing aims to propagate the modifications along the temporal dimension while maintaining overall structure and motion consistency with the source video, resulting in an edited target video $x^{tar} = \{x_i^{tar}\}_{i=1}^{L}$.

### 3.2 PRELIMINARIES

Flow-based generative models (Lipman et al., 2023; Liu et al., 2022) formulate a probability flow ODE over timestep $t \in [0, 1]$ to establish the transport map between the data distribution $p(x)$ and a standard Gaussian distribution $\mathcal{N} \sim (0, I)$:

$$\mathrm{d}z_t = v(z_t, t)\mathrm{d}t, \tag{1}$$

where $z_t$ stands for intermediate variables and $v$ is a time-dependent vector field usually parameterized by a neural network model. For the boundary condition, $z_1$ is the noise from $\mathcal{N} \sim (0, I)$, and $z_0$ is the data from $p(x)$. To generate a sample in $p(x)$, we initialize the ODE at $t = 1$ with a Gaussian sample $z_1$ and numerically solve the ODE backwards to obtain a sample $z_0$ that follows the distribution $p(x)$.

In practice, the ODE is solved numerically, with the timestep $t$ discretized into a sequence. Numerical ODE solvers are subject to discretization errors under curved ODE trajectories. Therefore, to encourage the trajectories to be *straight*, flow matching models typically learn the vector field with a linear interpolation between the noise and data, using the flow matching loss function:

$$\mathcal{L}_\theta = \mathbb{E}_{t, z_0 \sim p(x), z_1 \sim \mathcal{N}(0, I)} \left[ \|v_\theta(z_t, t) - (z_1 - z_0)\|^2 \right], \tag{2}$$

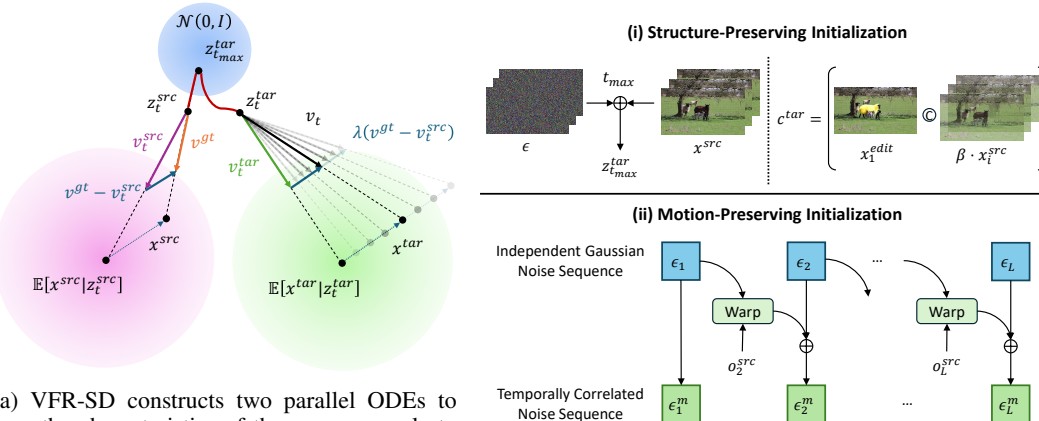

(a) VFR-SD constructs two parallel ODEs to map the characteristics of the source sample to the target distribution using a deviation term. The term is defined as the difference between the ground truth denoising vector $v^{gt}$ and the predicted source denoising vector $v_t^{src}$, and is used to rectify the target denoising vector $v_t^{tar}$.

(b) (i) Structural information of the source video is embedded through boundary condition initialization and flow model conditioning. (ii) Motion of the source video is incorporated through temporally correlated noise generated by warping with optical flow.

Figure 2: Illustration of VFR-SD (§3.3) and SMPI (§3.4).

where $\theta$ denotes the network parameters, and $z_t$ is a linear interpolation between $z_0$ and $z_1$:

$$z_t = (1 - t)z_0 + tz_1. \tag{3}$$

For the I2V task, the model takes an extra condition input $c$ to predict the vector field for the conditional distribution. $c$ includes the first frame of the generated video and the corresponding text prompt. For simplicity, we omit the text prompt and global conditions in $c$ in the following part.

## 3.3 VECTOR FIELD RECTIFICATION WITH SAMPLE DEVIATION (VFR-SD)

Given an image as the first frame condition, the I2V model can transform Gaussian noise into a video sample by solving an ODE according to the predicted vector field of the conditional distribution. When it comes to editing, we numerically solve the following ODE:

$$dz_t^{tar} = v(z_t^{tar}, t, c^{tar})dt, \tag{4}$$

so that the generated video $x^{tar} = z_0^{tar}$ is not only faithful to the edited frame $x_1^{edit}$ encoded in the target condition $c^{tar}$, but also consistent with the temporal evolution of original video $x^{src}$. However, the model only predicts a vector towards the *expectation* of the target distribution (Lipman et al., 2023; Gao et al., 2024; Chen et al., 2025), which hinders the preservation of sample-specific properties. As a prevailing method, inversion serves as a solution to incorporate sample-specific information by mapping $x^{src}$ to the initial Gaussian noise as the boundary condition $z_1^{tar}$. Nevertheless, this process is highly inaccurate and is often accompanied by model-specific designs to further inject information from the source video $x^{src}$ to ensure consistency.

To overcome the sample consistency challenge, VFR-SD exploits the probabilistic properties of flow-based models (Albergo & Vanden-Eijnden, 2023; Chen et al., 2025) by adding a sample-specific deviation to the target denoising vector. Specifically, while solving the ODE of the target video (Eq. (4)), VFR-SD also constructs a parallel ODE solving process for the source video:

$$dz_t^{src} = v(z_t^{src}, t, c^{src})dt. \tag{5}$$

As the source video is already given, we know the solution and the sample-specific ground truth vector field of Eq. (5). By leveraging the information along the ground truth denoising path of the source video, a target sample $x^{tar}$ can be produced while respecting the source sample without model-specific designs.

The core idea of VFR-SD is presented in Fig. 2a. Given the initial noise $z_{t_{max}}^{tar}$, we construct two parallel ODEs for the source and the target video distribution, respectively. The source latent variable

---

**ALGORITHM 1:** Vector Field Rectification with Sample Deviation (VFR- SD, §3.3)

---

**Input:** Source video $x^{src}$, source condition $c^{src}$, target condition $c^{tar}$, flow model $v_\theta$, initial timestep
  $t_{max}$, rectification scale $\lambda$.
**Output:** Edited video $x^{tar}$.
$\epsilon \sim \mathcal{N}(0, I)$
$z_{t_{max}}^{tar}, z_{t_{max}}^{src} \leftarrow (1 - t_{max})x^{src} + t_{max}\epsilon$ // Latents initialization.
$v^{gt} \leftarrow \epsilon - x^{src}$
// Numerically solve the parallel ODEs.
**for** $t \leftarrow t_{max}$ *downto* $0$ **do**
  $v_t^{tar} \leftarrow v_\theta(z_t^{tar}, t, \boldsymbol{c}^{tar})$ // Predict the target denoising vector.
  $v_t^{src} \leftarrow v_\theta(z_t^{src}, t, \boldsymbol{c}^{src})$ // Predict the source denoising vector.
  $v_t \leftarrow v_t^{tar} + \lambda(v^{gt} - v_t^{src})$ // Rectification.
  $z_{t-\Delta t}^{tar} \leftarrow \text{solver}_{t \rightarrow t-\Delta t}(z_t^{tar}, v_t)$ // Update target latents accordingly.
  $z_{t-\Delta t}^{src} \leftarrow \text{solver}_{t \rightarrow t-\Delta t}(z_t^{src}, v_t^{gt})$ // Update source latents with GT vector.
**end**
**return** $x^{tar} \leftarrow z_0^{tar}$

---

$z_t^{src}$ moves along the ground truth denoising path, while the target latent variable $z_t^{tar}$ is updated by the rectified denoising vector $v_t$. At each timestep $t$, the flow model predicts $v_t^{src}$ based on $z_t^{src}$, which points towards the expectation of the conditional source video distribution $\mathbb{E}[x^{src}|z_t^{src}]$. Then, we compute the difference between the ground truth denoising vector $v^{gt}$ and the model prediction $v_t^{src}$, which represents the sample-specific properties that deviate from the mean of the conditional distribution. Finally, we rectify the model-predicted target denoising vector $v_t^{tar}$ using the sample-specific deviation term above:

$$v_t = v_t^{tar} + \lambda(v^{gt} - v_t^{src}), \tag{6}$$

where $\lambda$ determines the rectification scale. The rectification term maps the deviation from expectation from the source distribution to the target distribution, thus preserving characteristics of the source video. Before the next iteration, the rectified vector $v_t$ is used to update the target latent $z_t^{tar}$ through an ODE solver. Meanwhile, the source latent variable $z_t^{src}$ is also updated using the ground truth denoising vector $v^{gt}$. We detail the complete algorithm in Alg. 1, in which the target condition $c^{tar}$ is the edited first frame $x_1^{edit}$ and the source condition $c^{src}$ is the original first frame $x_1^{src}$.

### 3.4 STRUCTURE-AND-MOTION-PRESERVING INITIALIZATION (SMPI)

To further preserve the structure and motion of the source video without modifying internal layers of the model, we designed SMPI (Fig. 2b) to incorporate such information into the boundary condition of ODE $z_{t_{max}}^{tar}$ and the target condition $c^{tar}$.

#### 3.4.1 STRUCTURE-PRESERVING INITIALIZATION

Previous research (Meng et al., 2022; Wang & Vastola, 2024; Liu et al., 2023; Hertz et al., 2023) suggests that visual outlines are generated in the early stages of diffusion sampling and details at later timesteps. Consequently, to enhance the structural information from the source video $x^{src}$, we select an initial timestep $t_{max}$ that is slightly smaller than the pure noise timestep $t = 1$, and initialize the boundary condition $z_{t_{max}}^{tar}$ as follows:

$$z_{t_{max}}^{tar} = (1 - t_{max})x^{src} + t_{max}\epsilon, \tag{7}$$

where $\epsilon$ is a sample from the standard Gaussian distribution. By exploiting the ground-truth denoising path of the source video in early steps, this strategy ensures a better consistency of the general layout between the source and the edited video.

For mainstream I2V models, the condition $c$ consists of the concatenation of the first frame and zero paddings to align with the video length $L$. We propose to leverage these unused paddings to encode information from the source video. Specifically, we compose the target condition $c^{tar}$ as follows:

$$c^{tar} = \text{concat}(x_1^{edit}, \beta\{x_i^{src}\}_{i=2}^L), \tag{8}$$

where $\beta$ is the embedding scale, which should be set to a small value to align with the training setting of the model. This approach allows further reference information from the source video.

### 3.4.2 MOTION-PRESERVING INITIALIZATION

There have been methods (Chang et al., 2024; Burgert et al., 2025) that replace the temporal Gaussianity with warped noise derived from optical flow to achieve motion control. However, these methods require extensive training. To ensure motion consistency with the source video in a training-free way, we devise a noise initialization strategy to encode the motion by temporal correlation while maintaining the general Gaussianity of the noise sample. Specifically, we first extract the optical flow of the source video $\{o_i^{src}\}_{i=2}^L$, and then modulate the independent Gaussian noise sequence $\epsilon = \{\epsilon_i\}_{i=1}^L$ with the motion cue as follows:

$$
\begin{aligned}
\epsilon_1^m &= \epsilon_1, \\
\epsilon_i^m &= \frac{1}{\sqrt{(1-\alpha)^2 + \alpha^2}}((1-\alpha) \cdot \texttt{warp}(\epsilon_{i-1}, o_i^{src}) + \alpha\epsilon_i),
\end{aligned}
\tag{9}
$$

where $\texttt{warp}$ stands for the 2D warping operation according to the optical flow, $\alpha$ is the blending factor to control the degree of temporal correlation, and the scaling factor is to preserve the unit covariance of the Gaussian distribution. The generated temporally correlated noise sequence $\epsilon^m = \{\epsilon_i^m\}_{i=1}^L$ is used to substitute the original i.i.d. noise sequence $\epsilon$ in Alg. 1 for enhanced motion prior.

### 3.5 DEVIATION CACHING (D-CACHE)

Calculating the rectification term $v^{gt} - v_t^{src}$ requires an additional pass through the flow-based model $v$, which almost doubles the computation. Inspired by recent work that explores using cached states to bypass some computation (Liu et al., 2025a; Zhao et al., 2025), we designed the D-Cache mechanism to reduce the cost of denoising vector rectification to an acceptable scale.

D-Cache reuses the previously calculated deviation term when the variation is small. Given that $v^{gt}$ is constant throughout the denoising process, it is $v_t^{src}$ that accounts for the variation. However, it is impossible to know the extent of its change before we compute $v_t^{src}$ through the model $v$. To estimate its change at the current timestep, we adopt the variation of the target denoising vector $v_t^{tar}$ as they are predicted by the same flow model $v$ to solve parallel ODEs at the same timestep $t$, and $v_t^{tar}$ can be obtained before predicting $v_t^{src}$. To be specific, we define the cumulative variation between $t_a$ and $t_b$ ($t_a < t_b$) as follows:

$$
d(t_a, t_b) = \sum_{t=t_a}^{t_b - \Delta t} \|v_t^{tar} - v_{t+\Delta t}^{tar}\|_1,
\tag{10}
$$

where $\Delta t$ is the step size. At the current timestep $t$, we calculate the cumulative variation starting from a previous timestep $t_p$. If $d(t, t_p) \leq \delta$, we use the cached source denoising vector $v_{t_p}^{src}$ as $v_t^{src}$ instead of predicting through the model. $\delta$ is designated as the caching threshold. By reusing the deviation term when the variation is minor, we achieve more efficient vector rectification without significantly compromising editing quality.

## 4 EXPERIMENTS

### 4.1 IMPLEMENTATION DETAILS

We select Wan2.1 (Wan et al., 2025) as the base I2V model to apply IF-V2V. It can generate 480p videos with 14B parameters. We adopt the Euler Discrete Scheduler (Esser et al., 2024) to solve the ODE with $t_{max} = 0.95$ and 25 sampling steps. Classifier-free guidance with scale 5.0 is applied when predicting the target denoising vector. The rectification scale $\lambda$ in §3.3 is set to 1.0. The embedding scale $\beta$ and the blending factor $\alpha$ in §3.4 are selected as 0.025 and 0.95, respectively. The caching threshold $\delta$ in §3.5 is set to 0.5. All other hyperparameters remain the same as Wan2.1 (Wan et al., 2025). Experiments are conducted on NVIDIA RTX 4090 GPUs.

### 4.2 QUALITATIVE RESULTS

We present various creative video editing results using IF-V2V in Figs. 1 and 3, including attribute modification (teaser, a, c.1, and d.1), object addition (teaser), object removal (b), and stylization (c.2

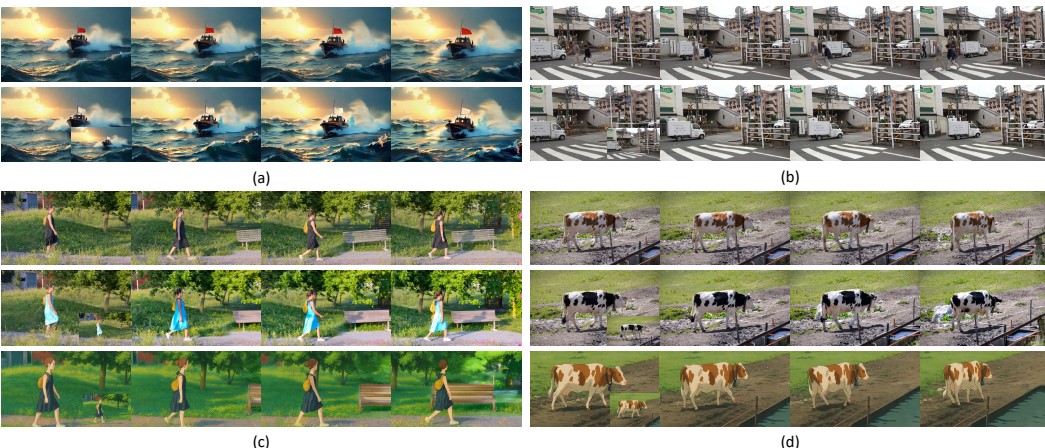

(a)                    (b)

(c)                    (d)

Figure 3: Editing results of IF-V2V (§4.2). In each case, the first row presents the original video, and the other rows show the edited video with the first frame condition in the bottom-right corner.

Table 1: Quantitative comparisons with previous methods (§4.3.1). Results in **bold** are the best. † Reference-based V2V without mask input.

| Method | AS | TC | EFC | HP | Time | Mem |
|---|---|---|---|---|---|---|
| Videoshop | $4.62 \pm 0.01$ | $97.87 \pm 0.02$ | $76.85 \pm 0.21$ | 1.69 | **96.67** | **26.18** |
| AnyV2V | $\underline{4.81} \pm 0.01$ | $97.88 \pm 0.02$ | $\underline{81.47} \pm 0.21$ | $\underline{2.56}$ | 1372.84 | 46.07 |
| VACE† | $4.57 \pm 0.01$ | $\underline{97.94} \pm 0.03$ | $75.65 \pm 0.20$ | 1.64 | 1603.10 | 46.83 |
| IF-V2V (Ours) | $\mathbf{4.88} \pm 0.01$ | $\mathbf{98.71} \pm 0.02$ | $\mathbf{92.79} \pm 0.06$ | **4.50** | $\underline{616.60}$ | $\underline{38.42}$ |

and d.2). As observed, IF-V2V achieves satisfying visual quality and consistency on a wide variety of image-conditioned video editing tasks thanks to the graceful collaboration between state-of-the-art image editing approaches (OpenAI, 2025; Liu et al., 2025b) and I2V models (Wan et al., 2025) empowered by our method. More results can be found in §G and the supplementary video.

## 4.3 COMPARISONS TO PRIOR WORKS

### 4.3.1 QUANTITATIVE COMPARISONS

To further demonstrate the superiority of IF-V2V over other methods, we quantitatively evaluate these approaches on 40 editing samples from the DAVIS (Perazzi et al., 2016) dataset and in-the-wild videos with a maximum of 81 frames. We construct these samples by editing the first frame of the video with GPT-4o (OpenAI, 2025) and Step1X-Edit (Liu et al., 2025b). We employ the following metrics to assess the editing quality and performance: 1) *Aesthetics Score (AS)* (Schuhmann et al., 2022): this metric evaluates the per-frame visual quality of the generated video. 2) *Temporal Consistency (TC)*: it assesses the smoothness of the edited video by calculating the average cosine similarity of CLIP (Radford et al., 2021) visual embeddings between every 2 consecutive frames. 3) *Edited Frame Consistency (EFC)*: it represents the consistency between the edited first frame and the generated video by the average cosine similarity of CLIP (Radford et al., 2021) visual embeddings. 4) *Human Preferences (HP)*: it stands for 13 volunteers' average rating on editing quality (5-point Likert Scale). 5) *Time*: it is the average time taken per video for the editing process in seconds. 6) *Mem*: it is the peak GPU memory consumption in gigabytes.

We compare IF-V2V with inversion-based methods, Videoshop (Fan et al., 2025b) and AnyV2V (Ku et al., 2024), and a training-based method, VACE (Jiang et al., 2025). For VACE, we compose the inputs as a reference-based V2V task without the mask input. As displayed in Tab. 1, IF-V2V consistently outperforms other approaches across all quality metrics, especially on EFC and HP. Compared to the inversion-based prior art AnyV2V (Ku et al., 2024), our method achieves consistently better results without inversion and model-specific design. Training-based method VACE (Jiang et al.,

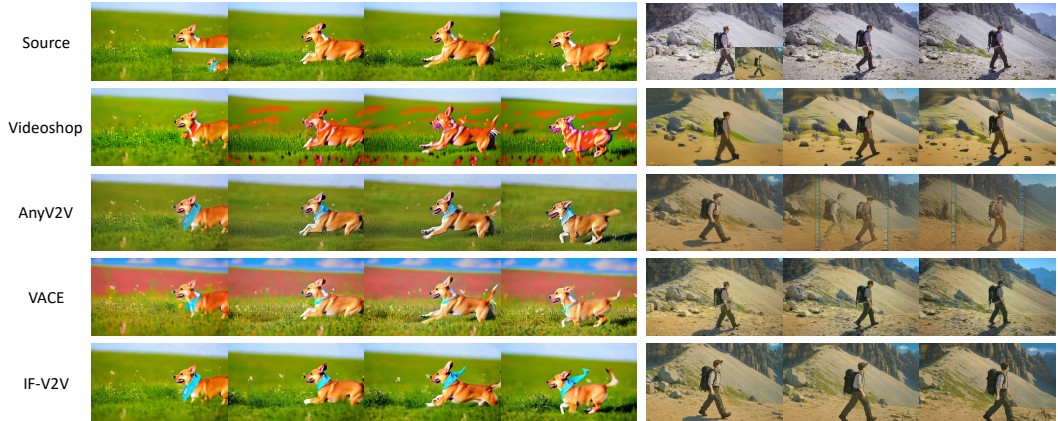

Figure 4: Qualitative comparisons with previous methods (§4.3.2). The edited first frame is in the bottom-right corner of the source video.

Table 2: Component ablations of IF-V2V (§4.4.1). **Bold** results are the best and underlined results are the second best.

| Setting | AS | TC | EFC | OVC | AEC | Time |
|---|---|---|---|---|---|---|
| I2V | $4.88 \pm 0.01$ | $98.70 \pm 0.03$ | $93.71 \pm 0.09$ | $75.03 \pm 0.22$ | $84.37 \pm 0.12$ | 554.27 |
| I2V + Init | $4.89 \pm 0.01$ | $98.30 \pm 0.02$ | $88.34 \pm 0.17$ | $78.74 \pm 0.20$ | $83.54 \pm 0.14$ | 553.52 |
| I2V + Inv | $4.79 \pm 0.01$ | $97.83 \pm 0.03$ | $86.87 \pm 0.14$ | $79.66 \pm 0.19$ | $83.26 \pm 0.14$ | 792.65 |
| *w/* VFR-SD | $4.77 \pm 0.01$ | $98.09 \pm 0.04$ | $92.50 \pm 0.10$ | $75.54 \pm 0.22$ | $84.02 \pm 0.12$ | 801.58 |
| *w/o* VFR-SD | $4.87 \pm 0.01$ | $98.29 \pm 0.03$ | $91.23 \pm 0.09$ | $75.27 \pm 0.22$ | $83.25 \pm 0.13$ | **553.58** |
| *w/o* SMPI | $4.78 \pm 0.01$ | $98.19 \pm 0.03$ | $92.67 \pm 0.08$ | $75.45 \pm 0.22$ | $84.06 \pm 0.12$ | 622.38 |
| *w/o* D-Cache | $4.87 \pm 0.01$ | $98.41 \pm 0.03$ | **$93.37 \pm 0.07$** | **$76.61 \pm 0.21$** | **$84.99 \pm 0.12$** | 804.46 |
| IF-V2V | **$4.88 \pm 0.01$** | **$98.71 \pm 0.02$** | $92.79 \pm 0.06$ | $76.44 \pm 0.20$ | $84.62 \pm 0.11$ | 616.60 |

2025) also falls behind our method when no editing mask is provided. As for the time and memory cost, IF-V2V outperforms all other methods except Videoshop (Fan et al., 2025b), which is based on a smaller I2V model (Blattmann et al., 2023) with less computational cost, but does not achieve satisfactory editing performance as demonstrated by AS, EFC, and HP.

### 4.3.2 QUALITATIVE COMPARISONS

We visualize the edited videos in Fig. 4 to provide an intuitive comparison with other methods. The left side shows an object addition task, where Videoshop and VACE exhibit significant artifacts. Although AnyV2V adds the blue scarf and preserves the dog's motion, the hue gets less vivid, and the background becomes blurry. Our method achieves the best result in inserting the blue scarf while maintaining the other aspects of the video. On the right side, we expect the models to alter the input video's style according to the given first frame. All the methods fail in this task except IF-V2V, further validating its effectiveness.

### 4.4 DIAGNOSTIC EXPERIMENTS

### 4.4.1 QUANTITATIVE ABLATIONS

To provide a better understanding of IF-V2V's components, we conduct ablation studies on the same editing samples as §4.3.1. Besides the objective metrics in §4.3.1, we additionally adopt the following metrics: 1) *Original Video Consistency (OVC)*: this metric measures the per-frame consistency between the edited video and the original video by the average cosine similarity of CLIP (Radford et al., 2021) visual embeddings. 2) *Average Editing Consistency (AEC)*: it is the mean value of EFC and OVC to assess the general editing consistency.

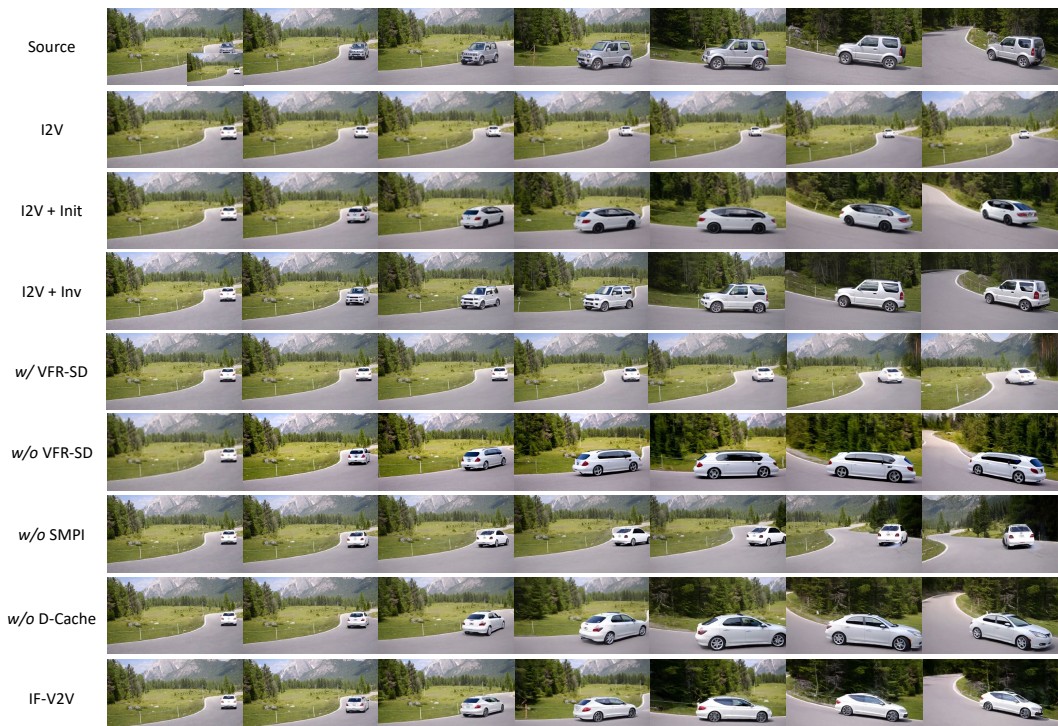

Figure 5: Case study for components (§4.4.2). We edit the first frame to be a white car with its **back towards the uphill direction**, expecting to generate a creative video in which the white car drives **backwards** up the hill. The edited first frame is in the bottom-right corner of the source video.

We present the quantitative results in Tab. 2. Three baseline methods are compared in the first three rows. I2V (#1) represents the result for directly adopting an I2V model (Wan et al., 2025). Although it achieves high TC and EFC, a large portion of the generated videos are *almost still*, which accounts for the high consistency scores. The OVC of #1 is also low because there is no information from the source video during generation. I2V + Init (#2) stands for using the I2V model (Wan et al., 2025) with initial latents generated by the linear combination of Gaussian noise and source video latents. Despite enhanced OVC, EFC significantly drops because information from the source video becomes dominant, and the model fails to integrate information from the edited frame. I2V + Inv (#3) represents using the ODE inversion technique to obtain the initial latents for I2V denoising. Despite a strong content preservation capability demonstrated by high OVC, the editing capability and result quality are not satisfactory according to other metrics. Moreover, the extra time cost is also pronounced.

#4 presents the results with only VFR-SD. Although VFR-SD achieves better content preservation as demonstrated by the enhanced OVC compared to #1, it does not perform satisfactorily without appropriate initialization of the ODE, especially in AS and TC. The time cost is also relatively high.

#5 demonstrates the results of If-V2V without VFR-SD. Despite the fastest inference time, EFC, OVC, and AEC are significantly behind those of IF-V2V (#8), demonstrating the capability of VFR-SD to incorporate characteristics of the source video while temporally propagating the edited frame.

#6 shows the metrics without SMPI. Compared to IF-V2V (#8), the drop of AS, TC, and OVC is relatively prominent. This validates SMPI's effect on better preserving the details in the original video to enhance the visual quality of the editing result.

#7 presents the results without the D-Cache mechanism. Despite slightly improved EFC, OVC, and AEC, the inference time increases significantly (**+30.5%**) compared to IF-V2V (#8), attesting to the acceleration effectiveness of D-Cache without notably compromising editing quality. Comparing #7 to #4 and #5, we can also confirm that both VFR-SD and SMPI are crucial for achieving satisfactory editing results, and their gains reinforce each other.

### 4.4.2 CASE STUDY

We further demonstrate the functions of IF-V2V's components with a creative editing sample in Fig. 5, which originally shows a grey van driving upwards a hill along the road. We edit the first frame to be a **white car** with its *back towards the uphill direction*, with an expectation of generating a creative video in which the white car drives *backwards* along the road up the hill.

If we directly generate the video with the I2V model (Wan et al., 2025) (#2), it fails to follow the text prompt, resulting in the white car driving forward down the hill. Initializing the latents with the source video as §4.4.1 (#3) does let the white car drives up the hill, but the generated car has obvious artifacts with *both sides being the back*. Meanwhile, this approach also suffers from inconsistent road shape and blurry output videos. Integrating inversion techniques (#4) cannot strike a satisfying balance between preserving the original video and propagating editing contents. Although the van becomes white, it does not turn into a car, and its heading direction is not successfully edited from the first frame.

#5 shows the results with only VFR-SD. Without proper initialization, VFR-SD cannot genuinely reproduce the motion from the source video. Results in #6 illustrate that without VFR-SD, the synthesized car also has two back ends. Meanwhile, there is a little corruption at the end of the road. The comparison between #6 and #8 displays that VFR-SD better preserves information from the source video, resulting in a more consistent and reasonable output. In #7, the white car first moves backwards for a little distance, then drifts towards the front right. Without SMPI, the method fails to preserve the motion from the source video. As observed, both VFR-SD and SMPI are essential for achieving the balance between the preservation of the original video content and the propagation of the edited first frame, and their improvements mutually enhance.

Both #8 and #9 successfully propagate the edited first frame to the source video to generate a white car driving backwards up the hill. We can observe that IF-V2V with the D-Cache mechanism offers an effective and user-friendly solution for creative editing with reasonable overhead.

## 5 CONCLUSION AND DISCUSSION

In this work, we propose IF-V2V, a user-friendly method to perform image-conditioned video editing by leveraging the strong temporal prior of pretrained flow-based I2V models. It includes VFR-SD to achieve inversion-free editing by introducing a deviation term into the denoising vector field to preserve source video information. SMPI is used to further enhance structure and motion consistency with the source video by embedding structural information into temporally related noise. D-Cache mechanism significantly reduces the additional computational cost, making IF-V2V more practical for common users. Extensive qualitative and quantitative results across various scenarios have validated the effectiveness of IF-V2V. We believe that IF-V2V will boost the creator's community by providing a handy tool to realize their creativity. More discussions are included in §H.

### REPRODUCIBILITY STATEMENT

The process of VFR-SD is detailed in Alg. 1, and all the hyperparameters of IF-V2V is listed in §4.1. We also discuss the theoretical justifications and hyperparameter selection criteria of IF-V2V in §H. Our code will be made publicly available to facilitate relevant research.

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

## APPENDIX OVERVIEW

The appendix includes extra experimental results, corresponding analyses, and further discussions of IF-V2V. The appendix is organized as follows:

- §A analyzes the effect of the rectification scale in §3.3.
- §B further ablates the components of SMPI.
- §C provides comparisons on adopting different ODE solvers.
- §D demonstrates quantitative and qualitative results of extending IF-V2V to other flow-based I2V models for editing.
- §E presents qualitative results of extending IF-V2V for text-guided video editing.
- §F adapts an inversion-free image editing method FlowEdit (Kulikov et al., 2024) to the video domain for quantitative comparison.
- §G shows more qualitative results of IF-V2V, including comparisons on longer videos.
- §H further discusses the theoretical justifications, hyperparameter selection criteria, limitations, and societal impacts of IF-V2V.

## A EFFECT OF THE RECTIFICATION SCALE

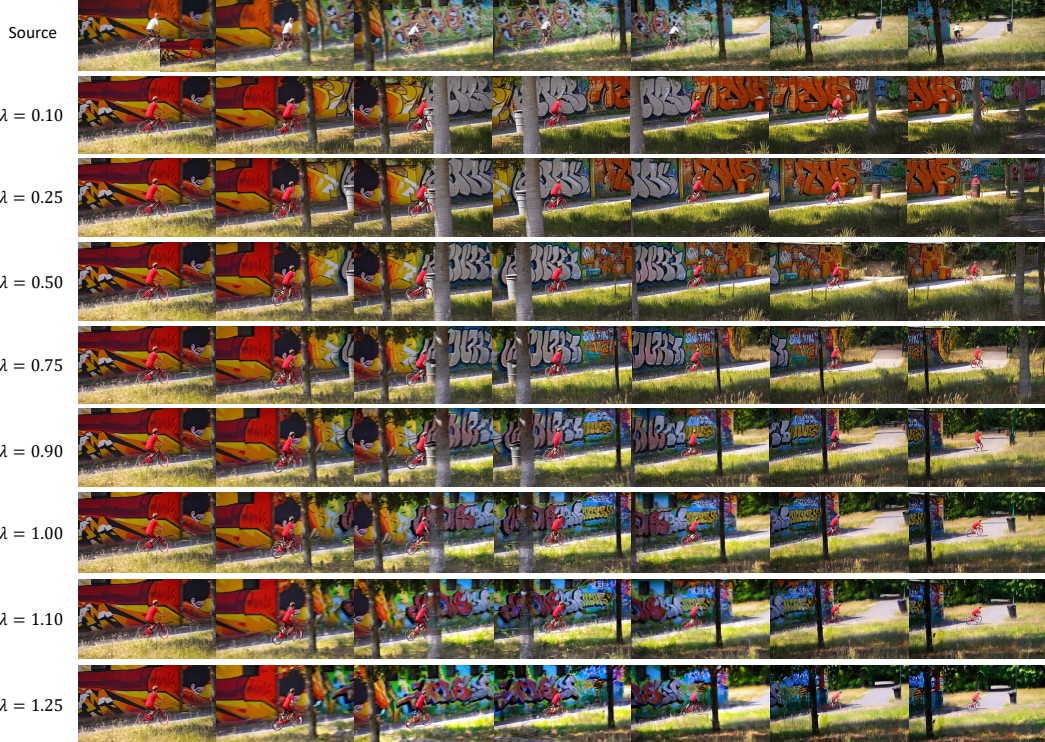

Figure 6: Illustration of the effect of the rectification scale $\lambda$ (§A). The source video shows a boy in a **white** T-shirt cycling along the road, and the edited first frame changes the boy's T-shirt to **red**. The value of $\lambda$ can be tuned in an appropriate range to control the strength of the source video prior. A small $\lambda$ ($\leq 0.50$) brings a weak prior from the source video, resulting in inconsistency with source video that the boy cycles along a road with an endless wall. When $\lambda$ is too large ($\geq 1.25$), artifacts like blurs and oversaturation also emerge. Please zoom in for details.

The rectification scale $\lambda$ in §3.3 determines the strength that VFR-SD incorporates the source video's deviation from expectation during the target ODE solving process. To provide an intuitive under-standing of the impact of $\lambda$ on the edited video, we present a visualization of using different values of $\lambda$ to edit a video sample in Fig. 6. The video originally captures a boy in a white T-shirt cy-cling along the road, and the edited frame turns the boy's T-shirt red. When $\lambda$ is small ($\leq 0.5$), the deviation from the source video is relatively weak, and the denoising process resembles the straight-forward I2V process. In this case, the sample deviation is inadequate to direct the denoising process,

resulting in the boy cycling along the road with an *endless* wall. In contrast, an overly large $\lambda$ value ($\geq$ 1.25) also induces artifacts like blurs and oversaturation because the rectification term pushes the generated sample too far from the original distribution.

## B    Extra Ablations on SMPI

We conduct extra ablations to provide a better understanding of SMPI. The results are displayed in Tab. 3, where *w/o* MPI stands for without motion-preserving initialization. Comparing #1 and #2, we can observe that structure-preserving initialization better maintains the consistency with original videos (OVC). From #2 and #3, it can be concluded that motion-preserving initialization further enhances temporal consistency.

Table 3: Quantitative ablations on SMPI (§B). Results in **bold** are the best.

| Setting | AS | TC | EFC | OVC | AEC | Time |
|---------|------|-------|-------|-------|-------|--------|
| *w/o* SMPI | 4.78 | 98.19 | 92.67 | 75.45 | 84.06 | 622.38 |
| *w/o* MPI | 4.81 | 98.06 | 92.51 | 76.30 | 84.41 | **615.17** |
| IF-V2V | **4.88** | **98.71** | **92.79** | **76.44** | **84.62** | 616.60 |

## C    Comparisons on ODE Solvers

To validate IF-V2V's compatibility with different ODE solvers, we evaluate IF-V2V's performance with Euler Discrete Scheduler (Esser et al., 2024) and UniPC Scheduler (Zhao et al., 2023). Quantitative results are displayed in Tab. 4 with the same settings as §4.4. We also present qualitative results in Fig. 7. From the above results, we can conclude that IF-V2V also achieves satisfactory performance with UniPC Scheduler (Zhao et al., 2023), demonstrating the universality of our method.

Table 4: Comparisons on adopting different ODE solvers in IF-V2V (§C).

| Setting | AS | TC | EFC | OVC | AEC |
|---------|------|-------|-------|-------|-------|
| UniPC (Zhao et al., 2023) | 4.86 | 98.70 | 92.01 | 76.48 | 84.25 |
| Euler (Esser et al., 2024) | 4.88 | 98.71 | 92.79 | 76.44 | 84.62 |

## D    Extension to Other Flow-based I2V Models

To further demonstrate the universality of IF-V2V, we select HunyuanVideo (Kong et al., 2025) as another base I2V model to apply our method. We present the quantitative results in Tab. 5, from which we can find that HunyuanVideo (Kong et al., 2025) also achieves a satisfying result that surpasses prior arts. Fig. 8 shows the visualization of some editing cases, in which we can observe that both HunyuanVideo (Kong et al., 2025) and Wan2.1 (Wan et al., 2025) achieve excellent consistency with both the edited frame and the original video with the help of IF-V2V.

Table 5: Comparisons on using different I2V models in IF-V2V (§D).

| Setting | AS | TC | EFC | OVC | AEC |
|---------|------|-------|-------|-------|-------|
| HunyuanVideo (Kong et al., 2025) | 4.75 | 98.60 | 92.92 | 75.36 | 84.14 |
| Wan2.1 (Wan et al., 2025) | 4.88 | 98.71 | 92.79 | 76.44 | 84.62 |

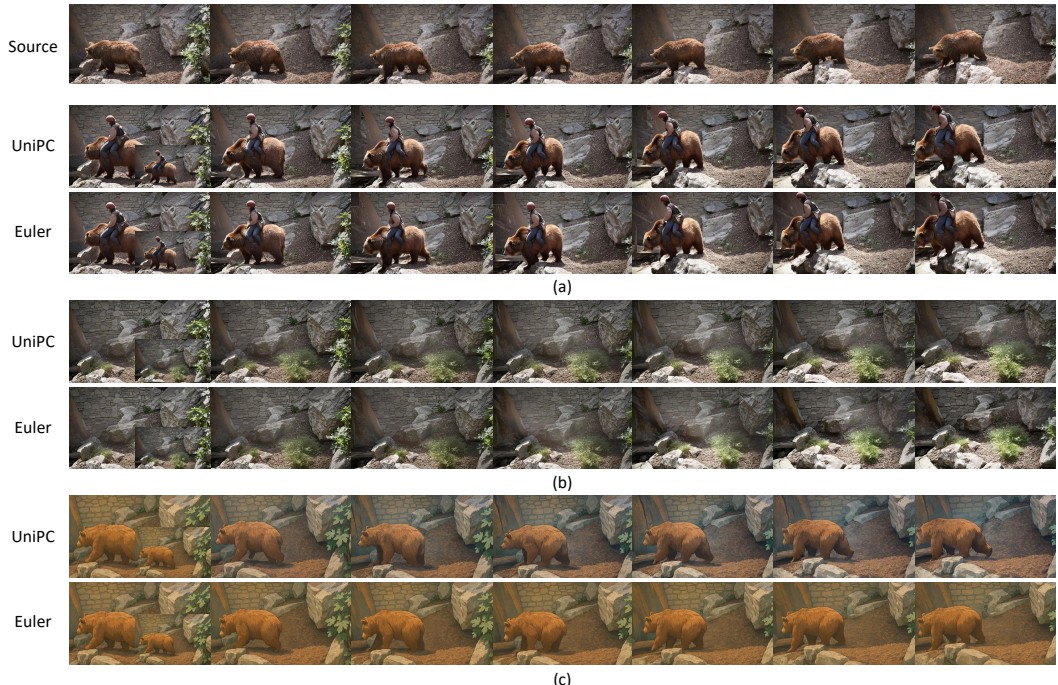

Figure 7: Visualizations of using different ODE solvers in IF-V2V (§C). The edited first frame is in the bottom-right corner of the edited video. IF-V2V is compatible with multiple ODE solvers to produce high-quality editing results.

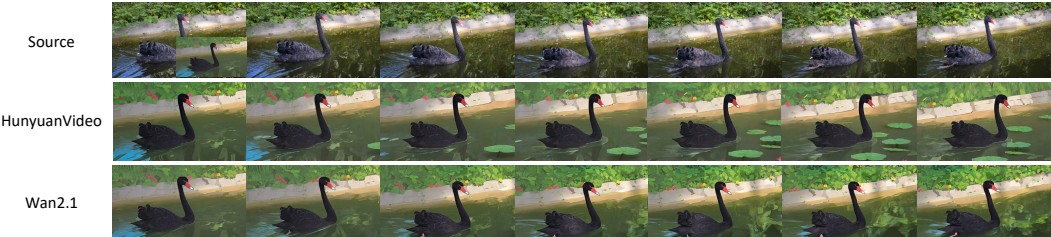

Figure 8: Editing samples of using different flow-based I2V models in IF-V2V (§D). The edited first frame is in the bottom-right corner of the source video. IF-V2V can be applied to various flow-based I2V models for high-quality video editing.

## E    EXTENSION TO TEXT-GUIDED EDITING

IF-V2V can also be used for text-guided video editing by removing the condition embedding in Structure-Preserving Initialization. We present some qualitative results using Wan2.1 (Wan et al., 2025) as the text-to-video model in Fig. 9, in which we can observe that IF-V2V also achieves excellent consistency and editing quality. In Fig. 9 (a), IF-V2V keeps the details more faithfully, such as shells on the beach and splashes in the sea, compared to directly blending Gaussian noise and source video latents as the initial condition for the I2V model (I2V + Init). In Fig. 9 (b), IF-V2V also aligns better with the editing prompt that alters the electric guitar into a normal one.

## F    QUANTITATIVE COMPARISONS WITH FLOWEDIT

To demonstrate the superiority of IF-V2V over directly adopting image-based inversion-free editing methods for videos, we adapt FlowEdit (Kulikov et al., 2024), a flow-based inversion-free image editing method, for video editing on Wan2.1 (Wan et al., 2025). The performance is displayed in

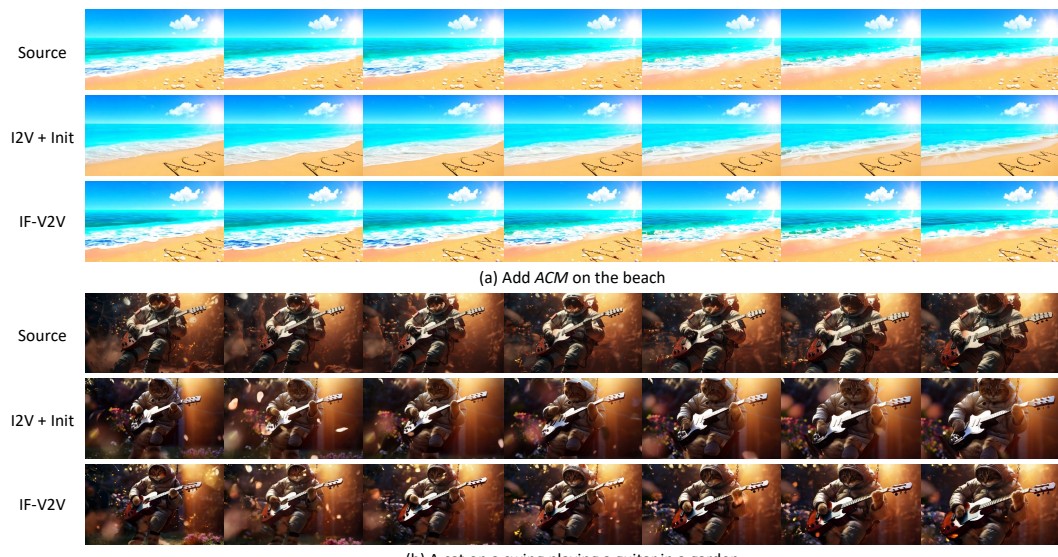

(a) Add *ACM* on the beach

(b) A cat on a swing playing a guitar in a garden

Figure 9: Text-guided editing results of IF-V2V (§E). We provide the simplified editing caption under each sample. Compared to *I2V + Init*, our method preserves the details more faithfully, like shells and splashes in (a), and aligns better with the editing instruction, such as the guitar in (b).

Table 6: Quantitative comparisons with FlowEdit (§F). Results in **bold** are the best.

| Setting | AS | TC | EFC | Time |
|---|---|---|---|---|
| FlowEdit (Kulikov et al., 2024) | 4.76 | 98.01 | 92.32 | 802.42 |
| IF-V2V (Ours) | **4.88** | **98.71** | **92.79** | **616.60** |

Tab. 6, from which we can observe that both its editing quality and inference speed remain inferior to IF-V2V. This further validates the effectiveness of our new perspective on the ODE solving process, video-specific designs, and flexible caching strategy.

## G    MORE VISUALIZATIONS

We illustrate more editing results of IF-V2V in Fig. 10, which include object addition (a), object removal (b), and attribute modification (c). As observed, IF-V2V consistently achieves satisfactory performance on various video editing tasks. Please refer to the supplementary video for dynamic versions of editing samples.

**Longer Videos.** To further demonstrate IF-V2V's compatibility with long videos, we implemented a sliding-window-based inference paradigm to support editing videos beyond the training length of the base I2V model. We edit two long video samples with 149 frames using such a paradigm, and adopt AnyV2V (Ku et al., 2024) to edit the same video samples downsampled to 81 frames (the maximum number of frames within the GPU memory limit). Visualizations are provided in Fig. 11, where AnyV2V shows significant temporal drifts and IF-V2V remains satisfactory consistency.

## H    MORE DISCUSSIONS

### H.1    THEORETICAL JUSTIFICATIONS

IF-V2V shares a similar theoretical basis as inversion-based editing methods: Optimal Transport (OT) mapping between the source and target distribution. The theoretical difference is that inversion-based methods conduct a mapping on the marginal distribution $p(z_0|z_{t_{max}})$, while IF-V2V performs mappings on transition distributions $p(z_{t-\Delta t}|z_t)$. When $\Delta t$ is small enough, both the source and tar-

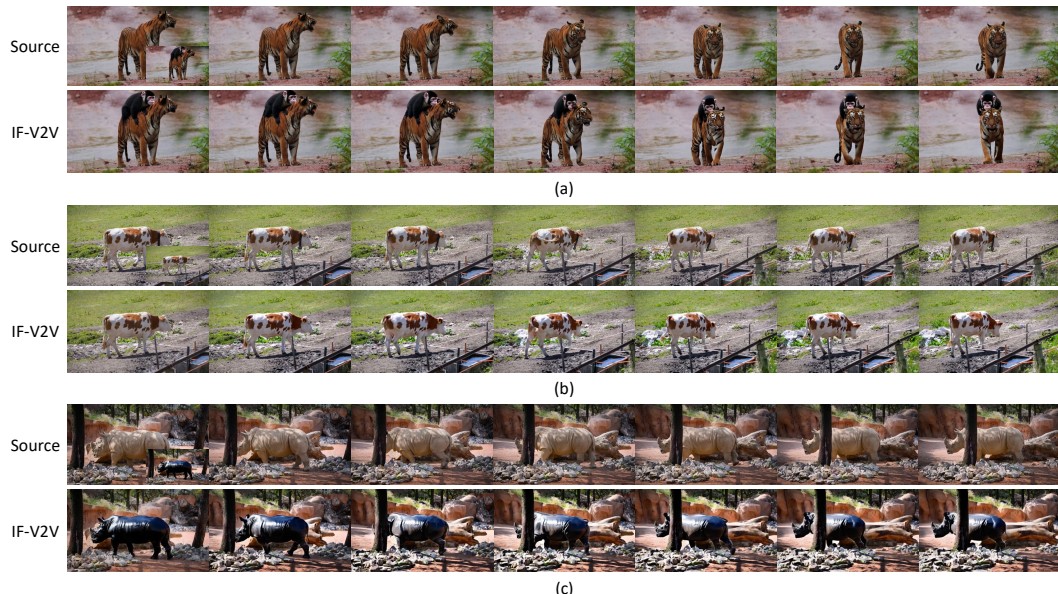

Figure 10: More visualizations of editing results of IF-V2V (§G). The edited first frame is in the bottom-right corner of the source video. The cases include object addition (a), object removal (b), and attribute modification (c). Our method propagates the edited frame to the whole video with excellent quality and consistency.

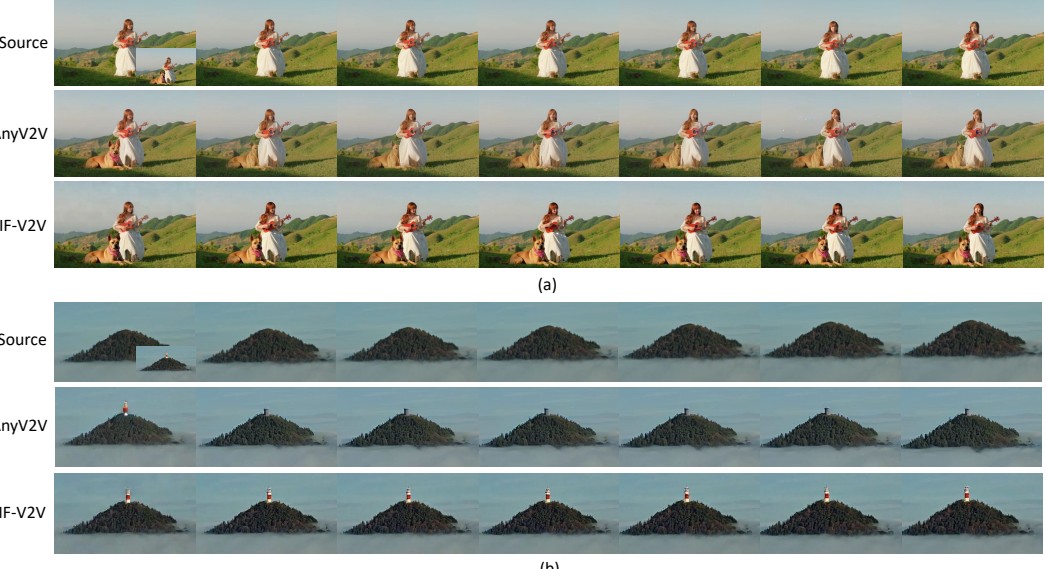

Figure 11: Visual comparisons of longer video editing results (§G). The edited first frame is in the bottom-right corner of the source video.

get transition distributions can be viewed as Gaussians with the same variance (Ho et al., 2020; Chen et al., 2025). In this case, IF-V2V with $\lambda = 1$ performs the exact OT mapping on the transitions.

## H.2 HYPERPARAMETER SELECTION

The editing results suffer from over-saturation and distortion when the rectification scale $\lambda$ in §3.3 is overly large. When the edited frame is not aligned with the original frame, the rectification some-

Table 7: Hyperparameter performance (§H.2). Results in **bold** are the best.

(a) AEC for different $\lambda$ values.

| $\lambda$ | 0.10 | 0.25 | 0.50 | 0.75 | 0.90 | 1.00 | 1.10 | 1.25 |
|---|---|---|---|---|---|---|---|---|
| AEC | 84.22 | 83.90 | 84.11 | 84.27 | 84.41 | **84.62** | 84.26 | 84.08 |

(b) AEC for different $\beta$ values.

| $\beta$ | 0.010 | 0.025 | 0.050 |
|---|---|---|---|
| AEC | 83.89 | **84.62** | 83.86 |

(c) AEC for different $\alpha$ values.

| $\alpha$ | 0.975 | 0.950 | 0.925 |
|---|---|---|---|
| AEC | 84.45 | **84.62** | 83.86 |

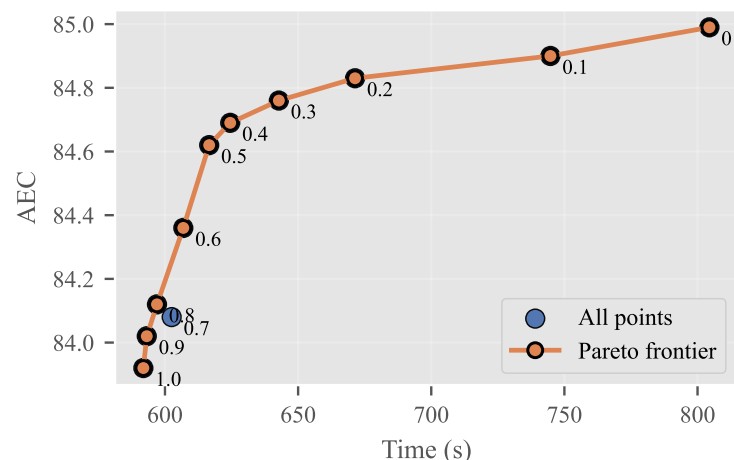

Figure 12: Pareto frontier for selecting $\delta$ (§H.2).

times causes unintended drifts due to the conflict. The scale of structure-preserving initialization $\beta$ in §3.4.1 should be small enough when the edited region is large. Otherwise, IF-V2V mostly preserves the original video. An overly large flow-guided initialization factor $\alpha$ in §3.4.2 breaks the temporal Gaussianity of the noise and fails the generation. We also provide the detailed hyperparameter performance in Tab. 7, where AEC is adopted as the main performance indicator for hyperparameter selection.

It has been discovered that initial steps are more crucial for editing, and the vector difference in these steps is also larger. The caching threshold $\delta$ in §3.5 is selected around the vector difference in early steps to avoid caching these steps. Caching in later steps reduces computational cost with less impact on editing quality. To facilitate practitioners to achieve an appropriate speed-quality balance, we also plot the Pareto frontier of $\delta$ (Time *vs.* AEC) in Fig. 12. We can observe that setting $\delta$ to around 0.5 achieves a satisfactory speed-quality balance at the elbow point of the curve.

## H.3 FIDELITY-CONSISTENCY TRADE-OFF

There is an inherent trade-off between the preservation of non-edited areas and the propagation of edited content in local editing cases. The propagation capability mainly comes from the generative capability of I2V models, while VFR-SD and SMPI mostly focus on preserving non-edited contents. The hyperparameters $\lambda$, $\beta$, and $\alpha$ control the extent of these content-preserving strengths. When the denoising vector prediction is noiseless, VFR-SD theoretically cancels out the alterations of non-edited areas when $\lambda = 1$. SMPI further preserves the original content by integrating original video latents into the ODE initialization process, given that the denoising vector prediction is inevitably inaccurate. With additional information from the source video, the target denoising path becomes more aligned with the source video, reducing the impact of inaccurate denoising vector prediction.

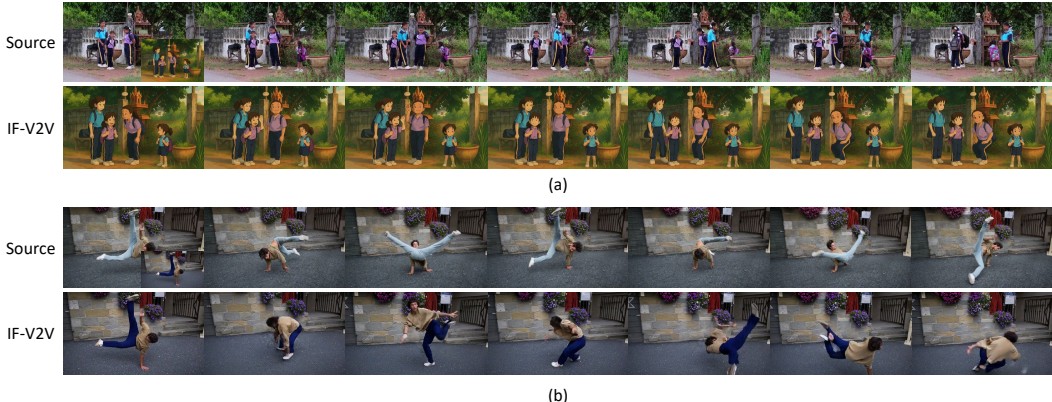

Figure 13: Failure cases of IF-V2V (§H.4.2). The edited first frame is in the bottom-right corner of the source video. IF-V2V struggles to handle editing samples with overly complex (a) or fast (b) motions due to the limited capability of I2V models.

## H.4 LIMITATIONS

The editing capability of our method is inherently bounded by the selected image editing model and the I2V model. Failure in either stage will result in unsatisfactory results. Moreover, since existing I2V models only predict the expectation of the distribution without covariance information, IF-V2V cannot exploit the covariance to achieve more precise mapping from the source sample to the target sample in a training-free way.

### H.4.1 IMAGE EDITING METHODS

IF-V2V's editing results rely on the first frame edited by image editing methods. However, current state-of-the-art methods (OpenAI, 2025; Liu et al., 2025b) still suffer from inconsistencies and trial-and-error. For instance, the image edited by GPT-4o (OpenAI, 2025) often misaligns with the original image, especially when changing the original image into a significantly different style (*e.g.*, Ghibli cartoonish style). Such misalignment may cause undesired alterations in the edited videos. In addition, Step1X-Edit (Liu et al., 2025b) sometimes needs several tries to achieve a satisfactory editing result. We expect that the future advancements of image editing methods will ease the process of obtaining a satisfactory first frame and further boost the performance of IF-V2V.

### H.4.2 I2V MODELS

IF-V2V fails to produce satisfactory results when motion in the source video is overly complex or fast. As Fig. 13 (a) displays, when there are complicated motions in the source video like simultaneous multiple subject movement with changing occlusions, IF-V2V cannot genuinely reproduce such motion in the edited video. In Fig. 13 (b), IF-V2V generates unsatisfactory results when dealing with breakdance, which contains rapid human body movements. These phenomena stem from state-of-the-art I2V models' limited ability to generate rapid or sophisticated motions. This problem may be resolved by more powerful I2V models in the future which are capable of handling such complex motions.

Furthermore, mainstream flow-based I2V models (Wan et al., 2025; Xu et al., 2024a; Kong et al., 2025; Yang et al., 2025; Peng et al., 2025; Fan et al., 2025a) only predict the *expectation* of the target distribution without further information like covariance, making it hard to conduct more fine-grained operations to map the source sample to the target distribution in a training-free way. This may limit the method's ability to maintain the consistency of fine-grained details in edited videos.

## H.5 SOCIETAL IMPACTS

IF-V2V can achieve high-quality video editing by combining off-the-shelf image editing and I2V methods without training, enabling practitioners to flexibly leverage the most up-to-date models to

implement their creativity. For individual creators, the lightweight nature of our method enables them to introduce AI-assisted video content creation into their workflow, democratizing the application of advanced AIGC tools. This shift can also expand storytelling beyond traditional media institutions to include diverse voices and perspectives. For commercial teams, our method provides them with a new chance to flexibly combine their internal results or models with the progress of the open-source community, boosting the quality of the produced videos with minor extra cost.

On the other hand, with IF-V2V's powerful capability of manipulating objects and attributes in the video, it can produce fabricated videos that appear highly realistic, posing significant challenges for verifying the authenticity of visual media. Such content can distort public perception and raise privacy concerns when fake contents featuring an individual are generated in an unauthorized way.

