# OpenReview forum: "Taming Flow-based I2V Models for Creative Video Editing"
_ICLR.cc/2026/Conference — Submitted to ICLR 2026_

### Official Review · Reviewer_JnVS · 2025-10-30

**Soundness:** 3
**Presentation:** 3
**Contribution:** 3
**Rating:** 6
**Confidence:** 3

**Summary:**

This paper presents IF-V2V, a novel inversion-free framework for video editing. The authors aim to address a key limitation in existing methods: the reliance on computationally expensive, model-specific inversion or optimization, which restricts the use of state-of-the-art, pre-trained Image-to-Video (I2V) models for editing tasks.

The core idea is to adapt off-the-shelf flow-matching-based I2V models without requiring an explicit inversion step. To achieve this, the method introduces several key technical contributions:

Vector Field Rectification with Sample Deviation (VFR-SD): A mechanism to incorporate information from the source video (e.g., non-edited regions) directly into the denoising vector field via a "deviation term," thus bypassing the need for inversion.

Structure-and-Motion-Preserving Initialization (SMPI): A model-agnostic noise initialization strategy designed to embed structural and motion cues from the source video into the initial noise, promoting temporal consistency and structural fidelity.

Deviation Caching (D-Cache): An efficiency-focused mechanism to minimize the additional computational overhead introduced by the vector field rectification process.

The authors claim that this lightweight, plug-and-play approach achieves superior editing quality and consistency compared to existing methods, offering a more flexible solution for video manipulation.

**Strengths:**

- The paper’s primary contribution lies in its highly practical approach. It strategically bypasses the significant challenges of training dedicated video editing models from scratch, such as the well-known scarcity of large-scale, paired video-text data and the computationally intensive, time-consuming training process. By proposing a framework designed to transfer the capabilities of established image editing models to the video domain, it offers a far more scalable, efficient, and data-agnostic solution.
- The introduction of VFR-SD and SMPI is a commendable aspect of this work. These components are well-motivated, as they directly address the critical challenge of maintaining consistency.
- The proposal of the Deviation Caching (D-Cache) mechanism to reduce the extra computational burden is a noteworthy contribution, enhancing the overall efficiency of the framework.

**Weaknesses:**

- Concerns remain regarding the editing consistency and fidelity. Upon inspection of Figure 3c (row 2), the later frames exhibit clear artifacts in non-edited regions, such as a noticeable increase in the saturation of the leaves. A similar issue is present in Figure 3d (row 2), where the final frame appears to suffer from semantic leakage originating from the edited subject. The authors should address these inconsistencies.

- Regarding Figure 8, while the method's applicability to diverse models is a clear strength, the results also highlight a potential weakness. The consistency of the edits across these different models appears limited. This variability is concerning, given that the editing objective is singular and explicit, and one would expect a more uniform outcome regardless of the underlying model.

**Questions:**

- The manuscript would be strengthened by a more in-depth discussion of the inherent trade-off between non-edited region fidelity and edit consistency. The authors should elaborate on the factors that control this balance and, specifically, how the method is designed to prevent semantic leakage from the edited subject or unintended alterations in static background regions.
- The paper's evaluation of temporal consistency for long-duration videos is insufficient. The current analysis does not provide a clear picture of how the method scales. A more rigorous analysis is needed to demonstrate that consistency can be maintained reliably over extended frame sequences.

---

> ### Author Response · Authors · 2025-11-21
> **Response to Reviewer JnVS**
>
> Thank you for recognizing IF-V2V as a highly practical approach with well-motivated designs and high efficiency. We also thank you for giving positive feedback. We would like to address your concerns as follows:
>
> **W1: Editing Consistency and Fidelity**
>
> VFR-SD theoretically depends on the accurate denoising vector prediction of the base I2V model. However, the base model's predictions are inevitably inaccurate since they are only approximate predictors, which may lead to undesired drifts during the editing process. This phenomenon can be mitigated by introducing masks to restrict the edited areas and keep the background unchanged. Nevertheless, this approach is only applicable to local edits and loses generalizability to global editing, such as style modification. Our method explores the balance between the generalizability of editing and the background preservation capability. Furthermore, the consistency of editing results will also improve with the advancement of the I2V model’s vector prediction accuracy.
>
> **W2: Consistency across Different Models**
>
> Although flow models are expected to predict the same denoising vector under identical conditions, they are only approximate predictors, and the prediction patterns are affected by the model architecture, data used, training paradigm, etc. Consequently, the behavior of different I2V models inevitably varies even under the same text and image conditions. IF-V2V performs rectification on the denoising vector predicted by flow-based I2V models, and its editing capability theoretically depends on the accuracy of the denoising vector prediction of flow models. With the advancement of flow-based I2V models, IF-V2V’s editing capability will be further enhanced by the improved accuracy of denoising vector prediction.
>
> **Q1: Trade-off between non-edited region fidelity and editing consistency**
>
> There is an inherent trade-off between the preservation of non-edited areas and the propagation of edited content in local editing cases. The propagation capability mainly comes from the generative capability of I2V models, while VFR-SD and SMPI mostly focus on preserving non-edited contents. The hyperparameters $\lambda$, $\beta$, and $\alpha$ control the extent of these content-preserving strengths (selection criteria detailed in §H.2). When the denoising vector prediction is noiseless, VFR-SD theoretically cancels out the alterations of non-edited areas when $\lambda=1$. SMPI further preserves the original content by integrating original video latents into the ODE initialization process, given that the denoising vector prediction is inevitably inaccurate. With additional information from the source video, the target denoising path becomes more aligned with the source video, reducing the impact of inaccurate denoising vector prediction. The discussion has been incorporated in the new §H.3.
>
> **Q2: Temporal Consistency for Long Videos**
>
> To further demonstrate IF-V2V’s compatibility with long videos, we implemented a sliding-window-based inference paradigm to support editing videos beyond the training length of the base I2V model. We edit two long video samples with 149 frames using such a paradigm, and adopt AnyV2V to edit the same video samples downsampled to 81 frames (the maximum number of frames within the GPU memory limit). Visualizations are provided in the new Figure 11, where AnyV2V shows significant temporal drifts and IF-V2V remains satisfactory consistency, demonstrating the editing capability of IF-V2V on longer videos. The discussion has been incorporated in §G.
>
> If you require more information, we are willing to supplement additional details during the discussion period. Thanks for your time and effort!

---

> ### Author Response · Authors · 2025-11-27
> **[Reviewer JnVS] Invitation to Participate in Discussions for ICLR Submission 2261**
>
> We want to express our sincere gratitude for your constructive feedback again, which recognizes IF-V2V as a highly practical approach with well-motivated designs and high efficiency. According to your suggestions, we have provided further clarifications and revised the manuscript. As the discussion period is nearing its end in just one week, we would like to invite you to join the discussion. If you require further clarification regarding your concerns, we are willing to supplement more details during the discussion period. We believe that an active discussion will be of great benefit to the authors, the reviewers, and the whole ICLR community.
>
> Thanks again for your time and effort!
>
> Best,
>
> ICLR Submission 2261 Authors

---

### Official Review · Reviewer_PtfT · 2025-10-31

**Soundness:** 2
**Presentation:** 3
**Contribution:** 2
**Rating:** 4
**Confidence:** 4

**Summary:**

The paper introduces IF-V2V, a training-free, inversion-free method that adapts flow-matching image-to-video models for image-conditioned video editing using an edited first frame. It runs two ODE trajectories in parallel (source and target) and adds a sample-specific deviation, which is the gap between the source’s ground-truth denoising vector and the model’s predicted one, onto the target’s vector field, preserving source details without explicit inversion. To boost fidelity and coherence, SMPI initializes latents with a structure-biased mix of source frames and noise, injects weak source-frame references into conditioning, and uses optical-flow-warped temporally correlated noise for motion. D-Cache reuses deviation terms when dynamics change slowly, lowering computation.

**Strengths:**

- **Training-free and inversion-free methods**
The approach is training-free, needs no further training or inversion calculation, and provides a simple way to utilize a pre-trained i2v model to achieve first-frame-guided video editing.

- **Model-agnosticism.**
The approach works well without a model-specific attention manipulation design. This could apply to a series of flow-matching-based models.

- **Resource efficiency.**
The D-Cache mechanism provides a way to reduce resource consumptions.

- **Various editing cases.**
The paper shows various editing cases, including object removal/insertion, stylization, etc, demonstrating its editing capability.

**Weaknesses:**

- **Ambiguity regarding model-agnosticism.**
The approach still needs manipulation on denoising vectors and the conditioning strategies. The padding-based structure injection depends on the specific conditioning layout of certain I2V models. This might not seamlessly transfer to all i2v models. The claim on  "agnosticism" is tricky. It achieves editing without attention manipulations, but it actually uses some other strategies that still need specific designs with the models.

- **Motion preservation has some issues**
According to the demo shown in the video, there are some issues in motion preservation (e,g, the walking man has a different body shape and direction from the source video). What is the reason behind this artifact? Is that because the VFR-SD cannot preserve well the source motions compared with inversion-based methods?

- **Comparisons**
Are the previous methods for comparison re-implemented on the same base models that the paper uses? If inversion plus attention manipulations are used to help preserve the source motions, will the performance be improved? A more in-depth analysis should be conducted to figure out if the current inversion-free methods are indeed better than inversion-based methods for the same model.

- **Runtime accounting is incomplete**
The paper does not provide runtime and resource consumption compared with other inversion-based methods and training-based methods.

**Questions:**

See "Weaknesses"

---

> ### Author Response · Authors · 2025-11-21
> **Response to Reviewer PtfT**
>
> Thank you for acknowledging IF-V2V’s training-free and inversion-free nature, model agnosticism, and resource efficiency. We would like to provide further clarifications as follows:
>
> **W1: Ambiguity regarding Model Agnosticism**
>
> We adopt the padding-based structure for the widely adopted state-of-the-art Wan model in the realm of video generation for enhanced structural preservation. All strategies except the padding-based structure injection are universally applicable to all flow-based I2V models, which take Gaussian noise latents as the initial input and predict the flow-based denoising vector to perform the transition from the Gaussian distribution to the target distribution. Even without such padding-based injection, IF-V2V still achieves satisfactory results on other flow-based I2V models without padding-based conditions like HunyuanVideo, as discussed in §D.
>
> **W2: Motion Preservation**
>
> VFR-SD and motion-preserving initialization work better in pixel-aligned editing cases. For style editing that significantly alters the pixel-wise correlation (like the style editing case of the walking man on the right side of Figure 4), there may be some drift in motion. It is worth noting that style editing without pixel alignment is rather challenging, and inversion-based methods generally fail in such cases, as shown on the right side of Figure 4.
>
> **W3: Base Models and Comparisons with Inversion**
>
> VACE uses the same base model as IF-V2V, and we use the official implementations of Videoshop and AnyV2V, whose base models differ from IF-V2V’s. Attention manipulations are model-specific operations that lack universality to other base models, and hinder the adoption of modern attention acceleration techniques like FlashAttention. Without FlashAttention, the memory cost and inference time of Wan-2.1-I2V will become unacceptable. To better address your concern, we implemented an inversion-based baseline method on Wan2.1-I2V for editing and have provided the results in the new Table 2 and Figure 5. Despite a strong content preservation capability demonstrated by high OVC, the editing capability and result quality are not satisfactory according to other metrics. Moreover, the extra time cost is also pronounced.
>
> **W4: Runtime Accounting**
>
> We have added runtime comparisons and peak GPU memory statistics in Table 1 (the original Table 1a). It can be observed that our method outperforms all other methods except Videoshop, which does not achieve satisfactory editing performance as demonstrated by AS, EFC, and HP, despite its fast inference time and reduced GPU memory cost. Moreover, compared to inversion-based methods that need to store the intermediate attention maps or latents on the disk (like AnyV2V), IF-V2V does not need to store any temporary results on the disk.
>
> If you require further clarification, we are willing to supplement more details during the discussion period. Thanks for your time and effort!

---

> ### Author Response · Authors · 2025-11-27
> **[Reviewer PtfT] Invitation to Participate in Discussions for ICLR Submission 2261**
>
> We want to express our sincere gratitude for your constructive feedback acknowledging IF-V2V’s training-free and inversion-free nature, model agnosticism, and resource efficiency. According to your suggestions, we have provided further clarifications and revised the manuscript. As the discussion period is nearing its end in just one week, we would like to invite you to join the discussion. If you require further clarification regarding your concerns, we are willing to supplement more details during the discussion period. We believe that an active discussion will be of great benefit to the authors, the reviewers, and the whole ICLR community.
>
> Thanks again for your time and effort!
>
> Best,
>
> ICLR Submission 2261 Authors

---

### Official Review · Reviewer_NABK · 2025-11-01

**Soundness:** 3
**Presentation:** 3
**Contribution:** 3
**Rating:** 6
**Confidence:** 3

**Summary:**

This paper presents an efficient optimization and inversion free method for video editing with flow based image generation models. The authors propose a novel vector field rectification technique to circumvent costly inversion. Structure and motion preserving initialization is used to preserve characteristics of the source video.

**Strengths:**

The paper introduces a novel contribution in vector field rectification. The comparative study demonstrates superior performance. Comparison with FlowEdit in the Appendix is also useful to place this paper.

**Weaknesses:**

Background preservation seems to be lacking in many examples, eg. person cycling (shirt color change example), woman walking in park example. etc. What is causing this inconsistency? What can be done to fix it?

Since efficiency is a major contribution and motivation in this paper, Table 1 a) should include comparison of time taken to make the edit with the baselines.

More video examples of edits would be good.

**Questions:**

None

---

> ### Author Response · Authors · 2025-11-21
> **Response to Reviewer NABK**
>
> Thank you for acknowledging our contribution in vector field rectification and giving positive feedback. We address your concerns as follows:
>
> **W1: Background Preservation**
>
> VFR-SD theoretically depends on the accurate denoising vector prediction of the base I2V model. However, the base model's predictions are inevitably inaccurate since they are only approximate predictors. This may lead to undesired drifts during the editing process, especially in scenes where the base model falls short (e.g., scenes with detailed background patterns or even texts, like the person cycling case in Figure 6). This phenomenon can be mitigated by introducing masks to restrict the edited areas and keep the background unchanged. Nevertheless, this approach is only applicable to local edits and loses generalizability to global editing, such as style modification. Our method explores the balance between the generalizability of editing and the background preservation capability. Furthermore, the consistency of editing results will also improve with the advancement of the I2V model’s vector prediction accuracy.
>
> **W2: Comparison of Time with Baselines**
>
> We have added editing time comparisons in Table 1 (the original Table 1a). It can be observed that our method outperforms all other methods except Videoshop, which does not achieve satisfactory editing performance as demonstrated by AS, EFC, and HP, despite its fast inference time.
>
> **W3: More Video Examples**
>
> We have incorporated more video examples in the supplementary video and are building a project page to accommodate more results.
>
> Should you require more information, we are willing to provide further details during the discussion phase. Thanks for your time and effort!

---

> ### Author Response · Authors · 2025-11-27
> **[Reviewer NABK] Invitation to Participate in Discussions for ICLR Submission 2261**
>
> We want to express our sincere gratitude for your constructive feedback, which acknowledges our novel contribution in vector field rectification. According to your suggestions, we have provided further clarifications and revised the manuscript. As the discussion period is nearing its end in just one week, we would like to invite you to join the discussion. If you require further clarification regarding your concerns, we are willing to supplement more details during the discussion period. We believe that an active discussion will be of great benefit to the authors, the reviewers, and the whole ICLR community.
>
> Thanks again for your time and effort!
>
> Best,
>
> ICLR Submission 2261 Authors

---

### Official Review · Reviewer_sprE · 2025-11-03

**Soundness:** 3
**Presentation:** 3
**Contribution:** 2
**Rating:** 4
**Confidence:** 3

**Summary:**

This paper attacks the “first-frame-conditioned” video-editing problem: given a source video and one edited first frame, generate a new video that (i) keeps the original structure & motion and (ii) respects the edit. Prior work either pays the price of diffusion inversion (heavy, noisy) or engineers attention maps for every new I2V backbone (fragile). The authors propose IF-V2V, a training-free, inversion-free, plug-and-play wrapper for any flow-matching I2V model. Key ideas: (1) run two ODE trajectories (source & target) in parallel and rectify the target vector field with the ground-truth source residual (VFR-SD); (2) inject structure & motion priors into the initial noise and the conditioning frame (SMPI); (3) cache the residual when its variation is small (D-Cache). Extensive experiments on Wan2.1 and HunyuanVideo show consistent gains over Videoshop, AnyV2V and VACE in visual quality, temporal consistency and human rating.

**Strengths:**

- First systematic attempt to bring inversion-free editing into flow-based I2V models; introduces the notion of “sample-deviation field” to transport source-specific residuals to the target ODE.
- Rigorous exposition: derives VFR-SD under the linear-Gaussian transition assumption and shows λ=1 performs optimal-transport on transition distributions.
- Solid empirical protocol: 40 DAVIS + in-the-wild clips, 4 task types, 6 metrics, ablations for every component, cross-backbone (Wan2.1 & HunyuanVideo) and cross-solver (Euler & UniPC) tests.
- Democratizes high-quality video editing: practitioners can combine the latest image-editor + I2V backbone with zero retraining.

**Weaknesses:**

- The paper presents three appealing components (VFR-SD, SMPI, D-Cache), but their interaction is described only intuitively; a unified objective or gradient-level view that shows whether the corrections reinforce or cancel one another is not provided.
- Table 1b shows very small margins on AS and TC; without standard errors or p-values it is unclear whether the bold-highlighted improvements exceed sampling noise.
- D-Cache gives a 25 % speed-up, yet the same table records a slight EFC drop; a concise Pareto curve (time vs. score) that clarifies this trade-off is absent.
- λ, β, α and δ are reported only as single optimal values; the searched grids and selection criterion are not given, limiting reproducibility.

**Questions:**

-Could you provide a joint-ablation (2×2×2) that shows how the three components combine—i.e., are the gains additive, multiplicative, or occasionally conflicting?
-Please run paired t-tests across clips and report 95 % confidence intervals for AS/TC/EFC to confirm which bold numbers are statistically meaningful.
-Would you plot a short Pareto frontier (wall-clock time vs. EFC/OVC) for several δ values so practitioners can choose an appropriate speed-quality operating point?
-Can you share the full hyper-parameter grids and the exact criterion (best EFC, best AEC, etc.) used to select λ, β, α and δ?

---

> ### Author Response · Authors · 2025-11-21
> **Response to Reviewer sprE**
>
> Thank you for acknowledging IF-V2V as the first systematic attempt to bring inversion-free editing into flow-based I2V models. We also appreciate your affirmation of the rigorous exposition, solid empirical protocol, and cross-solver tests. Your concerns are addressed as follows:
>
> **W1 & Q1: Joint ablation that shows how the three components combine**
>
> Since D-Cache depends on the parallel ODEs in VFR-SD, the D-Cache-only setting is invalid, and the SMPI-only setting is equivalent to the w/o VFR-SD setting. We supplement a VFR-SD-only setting in Table 2 (the original Table 1b) to make it a full joint ablation. As observed, both VFR-SD and SMPI are crucial for achieving satisfactory editing results, and their gains reinforce each other. D-Cache can significantly accelerate the editing process without notably compromising editing quality.
>
> **W2 & Q2: 95% confidence intervals to confirm statistical meaningfulness**
>
> We report 95% confidence intervals for AS/TC/EFC in Tables 1 and 2 (the original Table 1), from which the improvements can be statistically confirmed.
>
> **W3 & Q3: Pareto Frontier for $\delta$ illustrating speed-quality balance**
>
> We plot the Pareto frontier of $\delta$ (Time vs AEC, which is the mean value of EFC and OVC) in the new Figure 12 of the appendix. We can observe that setting $\delta$ to around 0.5 achieves a satisfactory speed-quality balance at the elbow point of the curve.
>
> **W4 & Q4: Hyperparameter selection criteria**
>
> We have provided the intuitive hyperparameter selection criteria in §H.2. To better address your concern, we further provide the AEC performance with different $\lambda$, $\beta$, and $\alpha$ values, respectively, in the new Table 7 of the appendix. AEC serves as the main performance indicator for hyperparameter selection. The selection criterion of $\delta$ has been explained in **Pareto Frontier for $\delta$**.
>
> If you have further questions, we are happy to provide further clarifications during the discussion phase. Thanks for your time and effort!

---

> ### Author Response · Authors · 2025-11-27
> **[Reviewer sprE] Invitation to Participate in Discussions for ICLR Submission 2261**
>
> We want to express our sincere gratitude for your constructive feedback, which acknowledges IF-V2V as the first systematic attempt to bring inversion-free editing into flow-based I2V models. We also appreciate your affirmation of the rigorous exposition, solid empirical protocol, and cross-solver tests. According to your suggestions, we have provided further clarifications and revised the manuscript. As the discussion period is nearing its end in just one week, we would like to invite you to join the discussion. If you require further clarification regarding your concerns, we are willing to supplement more details during the discussion period. We believe that an active discussion will be of great benefit to the authors, the reviewers, and the whole ICLR community.
>
> Thanks again for your time and effort!
>
> Best,
>
> ICLR Submission 2261 Authors

---

### Author Response · Authors · 2025-12-02
**Summary of the Reviews and Responses**

Dear Area Chairs,

We express our sincere gratitude for your devotion to the ICLR 2026 conference. To facilitate your evaluation, we would like to provide a brief summary of each reviewer’s comments and our responses as follows.

**Strengths**

Reviewers recognize that our work is the **first systematic attempt to bring inversion-free editing into flow-based I2V models** by rectifying the denoising vector field, with rigorous exposition, solid empirical protocol, and superior comparative performance. VFR-SD and SMPI are acknowledged as **well-motivated** components, as they directly address the critical challenge of maintaining consistency. Reviewers also appreciate our **model-agnosticism** and **resource efficiency**. Our method is completely training-free and inversion-free, **reaching a balance** among editing flexibility (both global and local), preservation of the original video, and resource efficiency.

**Concerns and Responses**

Reviewer sprE raised several **experiment-related concerns**, including the joint ablation that shows how the three components combine, 95% confidence intervals to confirm statistical meaningfulness, Pareto Frontier for $\delta$ illustrating speed-quality balance, and hyperparameter selection criteria. **All the experiments and the corresponding analysis have been supplemented.**

Reviewer NABK concerns **background inconsistency** in some samples, **comparison of time with baselines**, and **more video samples**. We provided **theoretical clarifications** regarding background inconsistency (the inherent inaccuracy of flow models), and potential solutions for local editing cases. We also provided **comparisons of time with baselines** and **more video samples**.

For reviewer PtfT, 1) we clarified our **model-agnostic essence** and referred to **results in the original paper on other flow-based I2V models** like HunyuanVideo as evidence. 2) For **motion preservation** problems, we explained that there may be some drift in **non-pixel-aligned cases** like style editing. For comparison, previous inversion-based methods generally fail in such cases. 3) For **base models and comparisons**, we clarified the base models for previous methods, explained **their lack of universality**, and implemented an **inversion-based baseline on the same base model** for comparison. 4) For **runtime accounting**, we have added **runtime comparisons** and **peak GPU memory statistics**.

For reviewer JnVS, we provided detailed explanations regarding **editing consistency and fidelity**, as well as **consistency across different base models**, where the **approximation nature of flow predictors** is the main cause. According to the reviewer’s suggestion, we also incorporated discussions on the **trade-off between non-edited region fidelity and editing consistency**, and qualitatively evaluated the **temporal consistency for long videos**.

The detailed response can be found in the official comment to reviewers. We have also revised the manuscript with the modified part highlighted in blue. We hope that the above summary can facilitate your decision-making process and make ICLR 2026 another success. Thanks again for your time and effort!

Best,

ICLR Submission 2261 Authors

---

### Meta-Review · Area_Chair_BX6f · 2026-01-02

**Summary:**

The paper presents an inversion-free method that is based on combination of the ground truth denoising trajectory of the source video with predicted denoising trajectory of the target video. The paper also presents structure and motion preserving initialization which are largely heuristic based approaches.

The main concern according to area chair was regarding quality of the generation (Background preservation (NABK), Motion preservation (PtfT), editing consistency and fidelity (JnVS)). This concern remains largely un-addressed during the discussion. Area chair carefully inspected the provided supplementary video and share the same concern.

Another large concern in area chair opinion that remain largely un-addressed was regarding comparison with state-of-the-art methods (Comparisons PtfT). Videoshop and AnyV2V was using much weaker base model, while VACE was used in V2V setting. None of these baselines utilize first frame obtained from image editing model, putting them at significant disadvantage compared to the proposed method. For the proposed method main heavy lifting was done by image editing model essentially, and edits just need to be propagated.

Given these 2 major concerns, area chair recommend rejection.

**Reviewer Concerns:**

The concerns of sprE was largely addressed by the authors.

Concerns of NABK, PtfT, JnVS regarding quality remains largely un-addressed.

Concerns NABK, PtfT regarding additional runtime estimates was addressed in area chair opinion.

Comparison weakness of PtfT was un-addressed, see above.

Applications to different models concern of (PtfT and JnVS) was relatively minor in area chair opinion and are reasonably addressed.

**Reviewer Scores:**

sprE - increase to 5-6

NABK - stay the same

PtfT - stay the same

JnVS - stay the same

---

### Decision · Program_Chairs · 2026-01-26

Reject